# Assessing the COVID-19 legacy on hand hygiene: Retrospective observational before–after study of compliance and alcohol-based

Amanda Carina Coelho de Morais[1]*, Sílvia Maria dos Santos Saalfeld[1],
Arthur Ricachenevsky[2], César Helbel[1], Matheus Cordeiro Marchiotti[1],
Hilton Vizi Martinez[1], Josy Anne Silva[1], Arthur Arenas Périco[2],
Fernanda Cristina Coelho Musse[2], Sanderland José Tavares Gurgel[2],
Jorge Juarez Vieira Teixeira[3], Maria Cristina Bronharo Tognim[3]

1 Hospital Infection Control Service, State University of Maringá, Maringá, Paraná, Brazil, 2 Department of Medicine, State University of Maringá, Maringá, Paraná, Brazil, 3 Department of Basic Health Sciences, State University of Maringá, Maringá, Paraná, Brazil

* amandacoelho_med@hotmail.com

## Abstract

In the context of multidrug-resistant organisms and following the coronavirus disease 2019 (COVID-19) pandemic, preventing healthcare-associated infections (HAIs) remains a major global public health priority. Although hand hygiene (HH) adherence increased markedly during the acute phase of the pandemic, there is limited empirical evidence on the sustainability and contextual variability of these behaviors once emergency conditions subside and routine clinical practice resumes. We conducted a retrospective observational before-and-after study to evaluate hand hygiene compliance (HHC) and HH product use among healthcare professionals working in adult, pediatric, and neonatal intensive care units (ICUs) of a teaching hospital, comparing pre-pandemic (09/01/2017–08/01/2018) and extended post-pandemic (10/29/2021–12/27/2024) periods. In the post-pandemic period, we also examined adherence across the World Health Organization's "Five Moments for Hand Hygiene." A total of 2,789 HH opportunities were recorded (1,048 pre-pandemic and 1,741 post-pandemic). Overall compliance increased from 61% before the pandemic to 66% in the post-pandemic period (p = 0.004). Compliance remained lowest during moments preceding patient contact and aseptic procedures, particularly in adult and pediatric ICUs, while the neonatal ICU consistently demonstrated higher performance. In parallel, HH practices shifted substantially, with alcohol-based hand rub use increasing more than fourfold compared with the pre-pandemic period (OR 4.30; 95% CI 3.32–5.58; p < 0.001). These findings indicate that crisis-driven improvements in HH were only partially sustained after the COVID-19 pandemic and varied across clinical contexts and HH moments. This variability underscores the need for continuous, context-sensitive infection-prevention strategies to support durable HH compliance beyond emergency situations.

**Data availability statement:** All data are in the manuscript and/or Supporting information files.

**Funding:** The authors received no specific funding for this work.

**Competing interests:** The authors have declared that no competing interests exist.

## Introduction

In the era of multidrug-resistant organisms and particularly following the coronavirus disease 2019 (COVID-19) pandemic, the prevention of healthcare-associated infections (HAIs) has become a critical global health priority [1–4]. Despite imprecise estimates, available evidence indicates that millions of individuals worldwide experience preventable harm or death due to failures in healthcare delivery, underscoring patient safety as a major public health concern [5,6]. In Europe, approximately 3.2 million patients are affected by HAIs annually, resulting in nearly 37,000 deaths, while in the United States about one in 25 hospitalized intensive care unit (ICU) patients acquires an HAI [7,8].

Within this context, hand hygiene (HH) is widely recognized as the most effective single measure for preventing hospital-acquired infections [5], as many pathogens are transmitted via the contaminated hands of healthcare workers when HH is omitted, delayed, or inadequately performed [6,9].

Over the past two decades, hand hygiene compliance (HHC) has been promoted globally by the World Health Organization (WHO), notably through the "Multimodal Hand Hygiene Improvement Strategy" launched in 2005, which integrates system change, education, monitoring and feedback, workplace reminders, and institutional safety climate reinforcement [10]. This approach was further strengthened by the "Clean Care is Safer Care" initiative [11,12] and operationalized through the "My Five Moments for Hand Hygiene", which provides a standardized framework for teaching, auditing, and reporting HH behavior [6,13].

During the COVID-19 pandemic, HH compliance were further reinforced through intensified infection prevention and control measures, including strong recommendations for alcohol-based hand rubs (ABHR) as the preferred method for routine HH [13]. Beyond compliance itself, HH product choice – such as the use of soap and water versus ABHR – offers insight into workflow efficiency and institutional safety culture. Due to its rapid action, skin tolerability, and accessibility at the point of care, ABHR has become a cornerstone of modern HH practice and a key driver of system-level improvements in adherence [14,15].

Although increased HHC during periods of heightened risk, such as the acute phase of the COVID-19 pandemic, is well documented, less is known about the sustainability of these behavioral changes once emergency conditions subside and routine clinical practice resumes [16]. Evidence suggests that crisis-driven improvements are not automatically sustained and may attenuate over time as perceived risk and institutional vigilance decline [17,18]. However, empirical data assessing how HHC patterns, adherence to specific WHO moments, and HH product use evolve in the post-pandemic period remain limited.

In this context, we investigated HHC among healthcare professionals working in adult, pediatric, and neonatal ICUs in a teaching hospital, comparing pre-pandemic and extended post-pandemic periods. We assessed overall compliance, HH product choice, and differences across ICUs and WHO "Five Moments for Hand Hygiene." Rather than establishing causal relationships, this study aims to characterize the sustainability, variability, and contextual heterogeneity of HH practices following the

COVID-19 pandemic, providing evidence relevant to the design of durable infection-prevention strategies beyond crisis situations.

## Methods

We conducted a retrospective observational before-and-after study using routinely collected HH surveillance data from a public teaching hospital in southern Brazil. We aimed to characterize changes in HH behavior and product use over time, analyzing variations associated with pre- and post-pandemic surveillance windows. The hospital maintains a structured HH audit program coordinated by the Hospital Infection Control Service. Data were collected in three intensive care units: Adult, Pediatric and the Neonatal. These units were selected because they represent distinct clinical environments with varying patient profiles, staff compositions, and workflow characteristics, allowing comparative assessment of HH practices across settings with different operational demands.

The analysis was based on all HH opportunities identified across both surveillance windows, reflecting real-world clinical workflow rather than a predefined sample of professionals. The study compared HH practices across two predefined surveillance periods reflecting distinct operational contexts in the hospital. The pre-pandemic period corresponded to routine institutional monitoring conducted between September 1, 2017, and August 1, 2018, when hand hygiene audits were performed periodically according to the hospital's standard surveillance cycle. The post-pandemic period extended from October 29, 2021, to December 27, 2024, during which continuous monitoring was implemented following the hospital-wide reinforcement of infection-prevention measures associated with COVID-19.

The difference in duration between the two periods reflects institutional changes in the auditing process rather than differences introduced by the study team. Before the pandemic, surveillance followed a fixed annual schedule and produced aggregated data. After the implementation of COVID-19 preventive measures, auditing became more frequent and systematic, resulting in a larger volume of observed opportunities and allowing stratification by WHO moments. Because the objective of the study was descriptive and all analyses were based on proportional measures (HHC divided by observed opportunities), comparisons were performed using relative metrics. The unequal duration of the surveillance windows reflects institutional changes in auditing practices rather than study design decisions.

### Observation procedures

Data collection was performed by medical students, all of whom received standardized training based on the WHO Hand Hygiene Observation Manual. The training covered the recognition of HH opportunities, classification of WHO moments, correct technique, and standardized criteria for judging whether HH actions were adequate. Observers were subsequently supervised by infection-control nurses, who are responsible for the hospital's HH audit program and who verified adherence to the observation protocol.

To minimize the Hawthorne effect, observations were performed covertly, meaning that healthcare professionals were not informed that they were being monitored. Although interobserver agreement statistics (e.g., kappa) were not recorded historically, all observers underwent the same structured training and direct supervision, and this limitation is explicitly acknowledged in the manuscript.

Each observation session lasted approximately 30 minutes and was distributed across morning, afternoon, and night shifts, on different days of the week, and across all three intensive care units. During each session, the observer registered every available hand hygiene opportunity occurring in the field of view, along with the corresponding action (performed or missed), the WHO moment (post-pandemic period), the professional category, and the type of product used (liquid soap, chlorhexidine, or alcohol-based preparation). The study did not involve direct enrollment of individuals; instead, the units of observation were HH opportunities recorded during routine surveillance activities conducted in the hospital's ICUs. All healthcare workers present during the observation periods could contribute to HH opportunities, including physicians, nurses, nursing technicians, physiotherapists, pharmacists, laboratory technicians, radiology technicians, and medical or nursing students.

**Definition of hand hygiene opportunity**

In accordance with WHO guidelines, a HH opportunity was defined as any instance in which HH action was indicated based on the "My Five Moments for Hand Hygiene" framework, meaning that a single healthcare worker could contribute multiple opportunities within one observation session. For the pre-pandemic period, HH indications followed national standards aligned with WHO definitions. This definition allowed standardized assessment across professions and units, ensuring comparability between periods despite differences in surveillance granularity.

**Sampling strategy and sample size**

Because this was a retrospective observational study based on the hospital's routine HH surveillance program, no sampling procedure was applied. Instead of selecting a subset of observations, we included all HH opportunities recorded during the predefined pre-pandemic and post-pandemic surveillance windows. This comprehensive approach is consistent with the purpose of institutional audit programs, which aim to capture real-world workflow patterns rather than produce probability-based samples. The total volume of opportunities reflects the natural variability in clinical activity and auditing intensity across the two periods. Because compliance was analyzed as a proportion (HHC divided by HH opportunities), the full inclusion of available data increased precision and reduced sampling bias.

**Bias control**

Several strategies were implemented to reduce potential sources of bias inherent to observational studies of HH. To minimize the Hawthorne effect, observations were performed covertly, and healthcare workers were unaware that their actions were being monitored. This approach follows WHO recommendations for reducing artificial inflation of compliance rates during direct observation. To address observer-related bias, all observers received standardized training based on WHO guidelines and were supervised by experienced infection-control nurses.

**Statistical analysis**

All analyses were conducted using IBM SPSS Statistics, version v31 (IBM Corp., Armonk, NY, USA). HHC was calculated as the proportion of HH actions performed divided by the total number of observed opportunities. To compare compliance between the pre-pandemic and post-pandemic periods – overall, by intensive care unit, and by professional category – we used the z-test for two independent proportions. Differences in the distribution of HH products (liquid soap or chlorhexidine vs. alcohol-based preparations) were evaluated using the chi-square test of independence, and the strength of association was quantified using odds ratios (ORs) with corresponding 95% confidence intervals.

For post-pandemic comparisons of the WHO "Five Moments" across intensive care units, we applied Pearson's chi-square test, reporting Cramer's V as a measure of effect size. When significant global differences were identified, pairwise Fisher's exact tests with Holm-Bonferroni correction were used to adjust for multiple comparisons.

Temporal patterns in HHC and product use during the post-pandemic period were described using quarterly aggregations, moving averages, and percentage variation between consecutive periods. A significance level of $p < 0.05$ was adopted for all analyses.

**Ethical considerations**

This study is part of the institutional research project "Strategies for reducing hospital costs and optimising the use of antimicrobials: rapid detection of resistance genes, synergism, and pharmacokinetic/pharmacodynamic analysis", phase 1 (CAAE 63610816.0.0000.0104) and phase 2 (CAAE 47908021.9.0000.0104). The project was approved by the Permanent Research Ethics Committee Involving Human Beings of the State University of Maringá. The present analysis used only routinely collected, aggregated, and non-identifiable HH surveillance data, without any direct interaction with

healthcare workers or patients; therefore, informed consent was not required under local ethical guidelines. Data were accessed by the authors between 1 June 2025 and 1 August 2025.

## Results

A total of 2,789 HH opportunities were recorded, with 1,048 before the COVID-19 pandemic and 1,741 after (S1 Table). The overall HHC rate before the pandemic was 61% (640/1,048) and 66% (1,157/1,741) (S1–S2 Tables) in the post-pandemic period. Comparative analysis of HHC before and after the pandemic showed a statistically significant increase (p=0.004). When analyzing ICUs separately, a substantial increase in adherence was observed in the Adult ICU, from 49.54% (164/331) to 62.88% (581/924) (p<0.001), and in the Neonatal ICU, from 65.29% (237/363) to 79.49% (314/395) (p<0.001). In contrast, the Pediatric ICU showed a non-significant decrease in adherence after the pandemic, from 67.51% (239/354) to 62.10% (262/422) (p=0.115) (S1 and S3 Tables).

Considering professional categories, compliance remained stable among physicians (70.8% vs. 70.2%; p=0.895). There was a non-significant increase among nurses (62.78% vs. 66.6%; p=0.223) and nursing technicians (54.18% vs. 56.42%; p=0.578). Among medical and nursing students, a non-significant reduction in HHC was identified (67.12% vs. 63.3%; p=0.614). On the other hand, other healthcare professionals (including physiotherapists, pharmacists, and laboratory and radiology technicians) showed a statistically significant increase in HHC, from 61.78% to 75% (p=0.012) (S1 and S4 Tables).

Before the pandemic, the most frequently used products for HH were non-medicated liquid soap and chlorhexidine (87%), with only 13% of healthcare workers opting for ABHR. After the pandemic, a major shift was observed, with ABHR accounting for 39% of HH events (S1 and S5 Tables). This change was statistically significant (OR 4.30; 95% CI 3.32–5.58; p<0.001). These findings are summarized in Table 1.

Table 2 summarizes HHC after the pandemic according to the WHO's "Five Moments" [6]. Marked variations were observed across ICUs. The compliance was significantly higher in the Neonatal ICU in Moments 1 (81.51%) and 2 (81.58%) compared with the Adult ICU (50.38% and 50.98%) and the Pediatric ICU (53.40% and 43.86%), with highly

**Table 1. Comparison of Hand Hygiene Compliance Rate before and after the coronavirus 2019 (COVID-19) pandemic, according to professional category and preparation used.**

|  |  | Before the pandemic | After the pandemic | *p value* |
|---|---|---|---|---|
| *HM Opportunities* |  | 1048 | 1741 |  |
| *Rate of HHC (HH actions/opportunities)* |  |  |  |  |
|  | *Overall HH compliance rate* | 61% (640/1048) | 66% (1157/1741) | *0,004** |
|  | Adult ICU | 49,54% (164/331) | 62,88% (581/924) | *<0,001** |
|  | Pediatric ICU | 67,51% (239/354) | 62,10% (262/422) | *0,115* |
|  | Neonatal ICU | 65,29% (237/363) | 79,49% (314/395) | *< 0,001** |
|  | Doctors | 70,8% (114/161) | 70,2% (198/282) | *0,895* |
|  | Students | 67,12% (49/73) | 63,3% (57/90) | *0,614* |
|  | Nurses | 62,78% (194/309) | 66,6% (610/916) | *0,223* |
|  | Nursing Technicians | 54,18% (207/382) | 56,42% (145/257) | *0,578* |
|  | Other Professionals | 61,78% (76/123) | 75% (147/196) | *0,012** |
| *Preparation used for HH (Rate of HHC; HH actions/opportunities)* |  |  |  |  |
| Liquid soap † |  | 87% (557/640) | 61% (705/1157) | *<0,001** |
| Alcohol-based preparations |  | 13% (83/640) | 39% (452/1157) | *OR 4,30 (3,32–5,58)* |

P-values refer to the chi-square test (or z-test for two independent proportions, when applicable). For the comparison of preparation type (soap vs. alcohol), the odds ratio (OR) and 95% confidence interval (95% CI) were calculated. Differences were considered statistically significant when p<0.05. Significant values are indicated with an asterisk (*). †Non-medicated liquid soap or chlorhexidine; HHC: Hand Hygiene Compliance.

**Table 2. Hand Hygiene Compliance among Adult, Pediatric and Neonatal Intensive Care Units (ICUs) after the COVID-19 pandemic according to the WHO "5 Moments".**

| Rate of HHC (%) and HM actions/opportunities | | | | |
|---|---|---|---|---|
| | **Adult ICU** | **Pediatric ICU** | **Neonatal ICU** | ***p* value** |
| Moment 1 | 50,38% (132/262) | 53,40% (55/103) | 81,51% (97/119) | *<0,001** |
| Moment 2 | 50,98% (52/102) | 43,86% (25/57) | 81,58% (31/38) | *<0,001** |
| Moment 3 | 64,79% (92/142) | 64,62% (42/65) | 80,85% (38/47) | *0,1027* |
| Moment 4 | 80,80% (202/250) | 77,98% (85/109) | 86,55% (103/119) | *0,2230* |
| Moment 5 | 61,31% (103/168) | 62,50% (55/88) | 62,50% (45/72) | *0,9757* |
| Preparation used for HH (Rate of HHC; HM actions/opportunities) | | | | |
| Liquid soap † | 58,35% (339/581) | 63,74% (167/262) | 63,38% (199/314) | 0,193 |
| Alcohol-based | 41,65% (242/581) | 36,26% (95/262) | 36,62% (115/314) | |

The World Health Organization (WHO) "Five Moments for Hand Hygiene": (1) before touching the patient; (2) before performing a clean/aseptic procedure; (3) after risk of exposure to bodily fluids; (4) after touching the patient; (5) after touching surfaces near the patient. P-values refer to Pearson's chi-square test (independence). Significant values are indicated with an asterisk (*). The effect size (Cramer's V) indicates a moderate effect at Moments 1 and 2 and a very low effect for soap vs. alcohol; †Non-medicated liquid soap or chlorhexidine; HHC: Hand Hygiene Compliance.

significant differences (p < 0.001 for both comparisons). Pairwise comparisons (Fisher's exact test with Holm adjustment) confirmed that, in both moments, the compliance was significantly higher in the Neonatal ICU compared with Adult and Pediatric ICUs. No significant differences were found between the Adult and Pediatric ICUs.

In Moment 3, the compliance across the three ICUs ranged from 64.62% to 80.85%, with no statistically significant differences (p = 0.1027). In Moment 4, the compliance was globally high (77.98% to 86.55%), but again showed no significant difference (p = 0.2230). Finally, in Moment 5, the compliance was similar across ICUs, around 61% to 62.5%, with no significant difference (p = 0.9757). Regarding the type of product used for HH after the pandemic, no significant difference was found between ICUs in the proportion of liquid soap use (58.35%–63.74%) *versus* alcohol-based preparations (36.26%–41.65%) (p = 0.193).

A time-series analysis of HH opportunities, product use, and compliance revealed a significant variation across the post-pandemic surveillance period (Fig 1; S6–S8 Tables; S1–S2 Figs). The highest number of opportunities occurred in October–December 2021 (n = 458), followed by a secondary peak in July–September 2024 (n = 325). In contrast, the lowest observation volume was recorded in April–June 2023 (n = 15), indicating fluctuating surveillance intensity. Soap-based HH predominated during most quarters, except for January–March 2022 and October–December 2024, when ABHR surpassed soap. Compliance rates fluctuated over time, ranging from 53.7% (Apr–Jun 2024) to 87.8% (Apr–Jun 2022) (Fig 2; S9 Table). A marked increase was observed from the initial period (64.9% in Oct–Dec 2021) to the first semester of 2022, reaching the highest compliance during the quarter of April–June 2022. From mid-2022 onward, compliance showed alternating declines and partial recoveries, without a sustained upward trend. The moving average curve demonstrated stabilization between 65% and 75% after 2023, suggesting moderate but consistent adherence following the pandemic peak. Trend analysis indicated a positive slope in early 2022 (+20.1%) followed by successive negative variations throughout late 2022 and early 2023, with intermittent recovery episodes thereafter.

## Discussion

During the COVID-19 pandemic, hand hygiene compliance among healthcare professionals increased compared with the pre-pandemic period, likely reflecting strengthened infection-prevention measures, intensified training, expanded access to ABHR, and heightened awareness of cross-transmission risks [16–19]. However, evidence regarding post-pandemic HHC has been inconsistent, with some studies reporting no sustained improvement, potentially due to

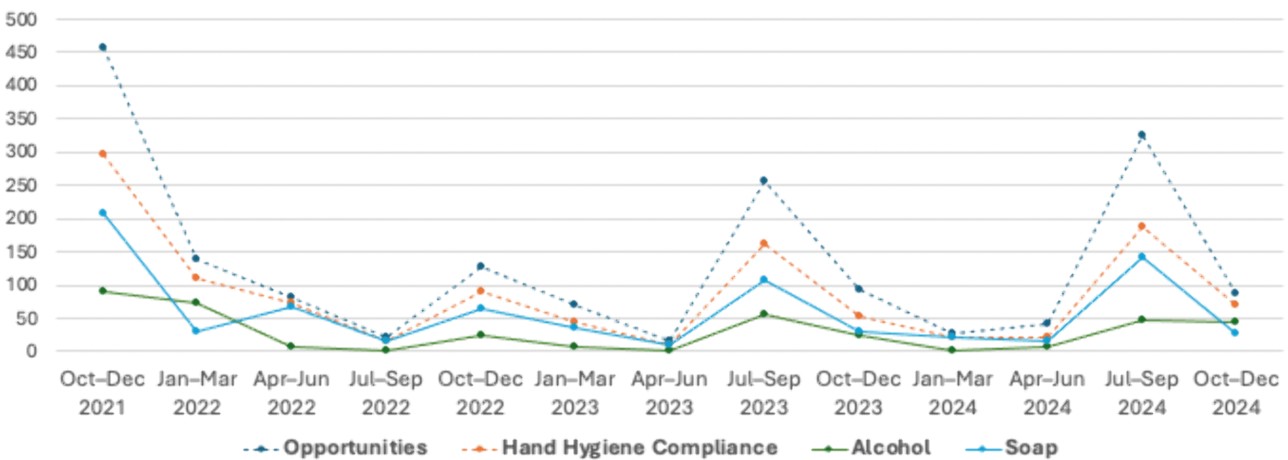

**Fig 1. Analysis of Hand Hygiene Opportunities, Compliance, and Use of Alcohol- or Soap-Based Formulations During the COVID-19 Pandemic.**

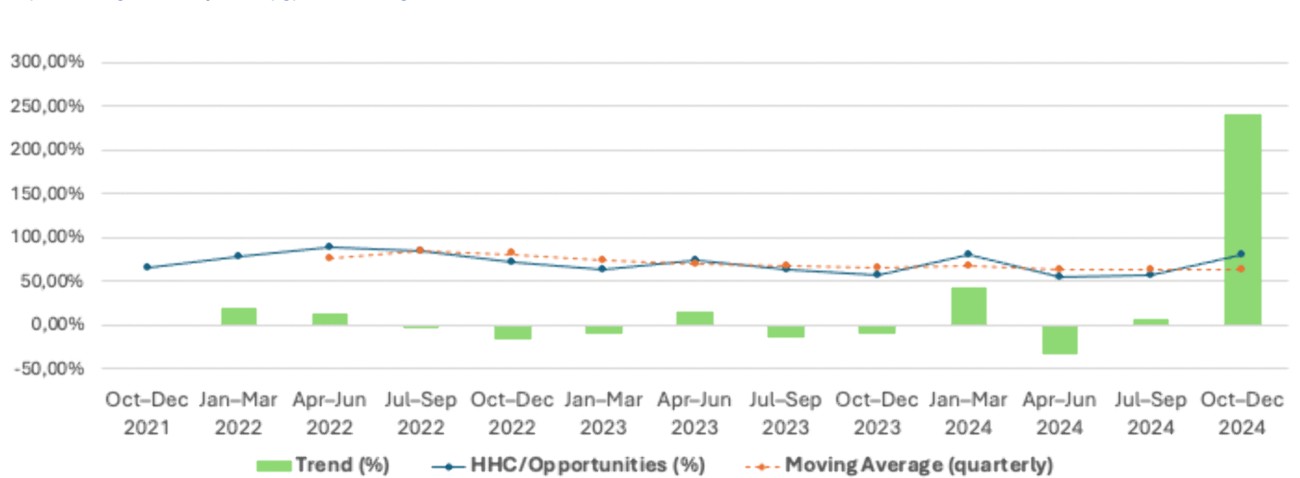

**Fig 2. Time-Series Analysis of HHC/Opportunities, Moving Average and Trend during the COVID-19 Pandemic.**

increased workload, workflow changes, or inappropriate substitution of hand hygiene by glove use [20–23]. In this study, we observed a statistically significant overall increase in HHC after the pandemic (p = 0.004), indicating partial persistence of crisis-driven behavioral changes.

Importantly, this persistence was not uniform across clinical contexts. Stratified analyses revealed significant post-pandemic increases in HHC in the Adult and Neonatal ICUs, whereas no significant change was observed in the Pediatric ICU. Higher compliance in the Neonatal ICU was particularly evident in the most critical moments of care and may reflect greater perceived vulnerability of newborns [21]. Conversely, lower risk perception in pediatric settings – supported by early pandemic evidence suggesting reduced susceptibility of children to severe COVID-19 – may have attenuated sustained adherence in that unit [21]. These findings highlight contextual heterogeneity as a key determinant of post-pandemic hand hygiene behavior.

Across professional categories, HHC remained largely stable, with significant improvement observed only among other healthcare professionals, such as physiotherapists, pharmacists, and laboratory and radiology technicians. Differences

between professional groups likely reflect variations in workload, task distribution, and workflow intensity, consistent with findings from multicenter studies [22].

Beyond compliance rates, our results demonstrate a marked transformation in hand hygiene behavior, reflected by changes in product use [23–26]. Prior to the pandemic, hand hygiene was predominantly performed using non-medicated liquid soap and chlorhexidine, with limited use of ABHR. In the post-pandemic period, ABHR use increased more than fourfold (OR 4.30; 95% CI 3.32–5.58), indicating substantial alignment with international recommendations and a structural shift in routine practice [27,28]. Because ABHR is faster, more accessible at the point of care, and better tolerated by healthcare workers, its increased use likely reflects not only improved availability but also maturation of institutional safety culture and workflow optimization [14,28].

Analysis of adherence across the WHO "Five Moments for Hand Hygiene" further reinforces the role of risk perception in shaping behavior. Higher compliance was consistently observed after patient contact or exposure, whereas lower adherence persisted in moments preceding patient contact or aseptic procedures, particularly in Adult and Pediatric ICUs. This pattern, reported in previous studies [17–19,26], suggests that immediate perceived risk continues to drive compliance more strongly than anticipatory prevention, even in the post-pandemic context.

Temporal analysis revealed substantial variability in observation volume and compliance over time, underscoring the influence of operational and contextual factors on surveillance data [29]. The lowest volume of hand hygiene observations occurred between January and April 2023, reaching a minimum in April–June 2023. This decline reflects a temporary reduction in observation capacity rather than a true decrease in compliance and was attributable to disruptions in the academic calendar following the pandemic, which limited the availability of trained observers [30]. The subsequent recovery of observation volume supports the interpretation that this fluctuation represents a surveillance artifact rather than a behavioral change [31].

Taken together, these findings directly address an important gap in the literature: while increased hand hygiene adherence during health crises is well established, less is known about how these behaviors evolve and persist once emergency conditions subside [32,33]. Our results show that post-pandemic sustainability of hand hygiene practices is partial, context-dependent, and accompanied by meaningful behavioral transformation, particularly in product choice. This heterogeneity underscores the need for continuous, context-sensitive infection-prevention strategies tailored to specific clinical environments, rather than reliance on crisis-driven vigilance alone [9,10].

This study has limitations. Direct observation is subject to observer bias, and interobserver agreement statistics were not available, although standardized training and supervision were applied. The single-center design limits generalizability, and the retrospective before-and-after design precludes causal inference. Unmeasured contextual factors, such as staffing patterns or local educational initiatives, may have influenced observed trends. Accordingly, our findings should be interpreted as descriptive evidence of post-pandemic behavioral patterns rather than causal effects.

## Conclusion

The results reinforce that the COVID-19 pandemic led to an increase in healthcare workers' hand hygiene compliance, although disparities persist among professional categories and hospital sectors. The use of alcohol-based hand rubs after the pandemic showed a significant increase, with more than a fourfold higher likelihood of alcohol use compared with the pre-pandemic period, in accordance with WHO recommendations. Overall, the findings indicate that HHC is more critical before patient contact or before aseptic procedures, particularly in Adult and Pediatric ICUs; whereas the Neonatal ICU demonstrates superior performance across all moments. The temporal fluctuations demonstrate that hand hygiene compliance is strongly influenced by contextual and operational factors, highlighting the ongoing need for continuous surveillance and focused educational interventions to ensure sustained and optimal performance.

**Supporting information**

**S1 Table. Opportunities for hand hygiene according to professional category and the World Health Organization (WHO) Five Moments for Hand Hygiene during the COVID-19 pandemic.** The table presents direct observational data collected in adult, pediatric, and neonatal intensive care units, stratified by date, hospital sector, and professional category. Hand hygiene opportunities were classified according to the WHO Five Moments: (1) before contact with the patient; (2) before performing an aseptic procedure; (3) after risk of exposure to bodily fluids; (4) after contact with the patient; and (5) after contact with surfaces near the patient. Abbreviations: S, hand hygiene performed; N, hand hygiene not performed; A, alcohol-based hand rub used; 0, no opportunity observed.
(DOCX)

**S2 Table. Opportunities for hand hygiene before the COVID-19 pandemic.** Aggregated data covering the period from January 9, 2017, to January 8, 2018, in adult, pediatric, and neonatal intensive care units.
(DOCX)

**S3 Table. Rate of hand hygiene compliance in adult, pediatric, and neonatal ICUs before the COVID-19 pandemic.** Compliance rates calculated as hand hygiene actions divided by observed opportunities.
(DOCX)

**S4 Table. Rate of hand hygiene compliance by occupational category before the COVID-19 pandemic.** Compliance rates stratified by professional category.
(DOCX)

**S5 Table. Hand hygiene compliance rate by product type before the COVID-19 pandemic.** Distribution of hand hygiene actions according to product used (liquid soap or alcohol-based preparation) and mean hand rubbing time.
(DOCX)

**S6 Table. Analysis of hand hygiene opportunities, compliance, and use of alcohol- or soap-based formulations during the COVID-19 pandemic.** Aggregated quarterly data summarizing total opportunities, number of hand hygiene compliance, and type of product used.
(DOCX)

**S7 Table. Time-series analysis of hand hygiene compliance during the COVID-19 pandemic.** Quarterly compliance data including moving average and trend estimates.
(DOCX)

**S8 Table. Time-series analysis of hand hygiene opportunities during the COVID-19 pandemic.** Quarterly number of observed opportunities with corresponding moving averages and trend estimates.
(DOCX)

**S9 Table. Time-series analysis of hand hygiene compliance rate (hand hygiene actions/ opportunities) during the COVID-19 pandemic.** Quarterly compliance rates expressed as percentages, including moving averages and trend estimates.
(DOCX)

**S1 Fig. Time-series analysis of hand hygiene compliance during the COVID-19 pandemic.**
(DOCX)

**S2 Fig. Time-series analysis of hand hygiene opportunities during the COVID-19 pandemic.**
(DOCX)

## Author contributions

**Conceptualization:** Amanda Carina Coelho de Morais, César Helbel, Maria Cristina Bronharo Tognim.

**Data curation:** Amanda Carina Coelho de Morais, Sílvia Maria dos Santos Saalfeld, César Helbel, Matheus Cordeiro Marchiotti, Hilton Vizi Martinez, Josy Anne Silva, Maria Cristina Bronharo Tognim.

**Formal analysis:** Amanda Carina Coelho de Morais, Sílvia Maria dos Santos Saalfeld, Fernanda Cristina Coelho Musse, Maria Cristina Bronharo Tognim.

**Investigation:** Amanda Carina Coelho de Morais, Sílvia Maria dos Santos Saalfeld, César Helbel, Hilton Vizi Martinez, Josy Anne Silva, Jorge Juarez Vieira Teixeira, Maria Cristina Bronharo Tognim.

**Methodology:** Amanda Carina Coelho de Morais, Fernanda Cristina Coelho Musse, Sanderland José Tavares Gurgel, Jorge Juarez Vieira Teixeira, Maria Cristina Bronharo Tognim.

**Project administration:** Amanda Carina Coelho de Morais, Hilton Vizi Martinez, Maria Cristina Bronharo Tognim.

**Resources:** Amanda Carina Coelho de Morais.

**Software:** Amanda Carina Coelho de Morais, Arthur Ricachenevsky, Arthur Arenas Périco.

**Supervision:** Maria Cristina Bronharo Tognim.

**Validation:** Amanda Carina Coelho de Morais, César Helbel, Matheus Cordeiro Marchiotti, Fernanda Cristina Coelho Musse, Sanderland José Tavares Gurgel, Maria Cristina Bronharo Tognim.

**Writing – original draft:** Amanda Carina Coelho de Morais, Fernanda Cristina Coelho Musse, Maria Cristina Bronharo Tognim.

**Writing – review & editing:** Amanda Carina Coelho de Morais, Arthur Ricachenevsky, César Helbel, Arthur Arenas Périco, Fernanda Cristina Coelho Musse, Sanderland José Tavares Gurgel, Jorge Juarez Vieira Teixeira, Maria Cristina Bronharo Tognim.

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
