## [Decision Letter · Decision Letter 0]

21 Oct 2025

PGPH-D-25-02547

HAND HYGIENE, COMPLIANCE AND ALCOHOL-BASED PRODUCTS: WHAT WAS THE LEGACY OF THE COVID-19 PANDEMIC?

Dear Dr. Coelho de Morais,

Thank you for submitting your manuscript to PLOS Global Public Health. After careful consideration, we feel that it has merit but does not fully meet PLOS Global Public Health’s publication criteria as it currently stands. Therefore, we invite you to submit a revised version of the manuscript that addresses the points raised during the review process.

Methodology of the study is not clear raising concerns on the quality and validity of the results. Comments and queries from the reviewers need to be clarified before further consideration.

We look forward to receiving your revised manuscript.

Kind regards,

Sonali Sarkar

Academic Editor

Journal Requirements:

1. Please provide a complete Data Availability Statement in the submission form, ensuring you include all necessary access information or a reason for why you are unable to make your data freely accessible. If your research concerns only data provided within your submission, please write "All data are in the manuscript and/or supporting information files" as your Data Availability Statement.

Additional Editor Comments (if provided):

The study is of public health importance, but lacks the methodological rigour.

Reviewers' comments:

Reviewer's Responses to Questions

**Comments to the Author**

1. Does this manuscript meet PLOS Global Public Health’s publication criteria?

Reviewer #1: Partly

Reviewer #2: Partly

2. Has the statistical analysis been performed appropriately and rigorously?

Reviewer #1: No

Reviewer #2: No

3. Have the authors made all data underlying the findings in their manuscript fully available (please refer to the Data Availability Statement at the start of the manuscript PDF file)?

Reviewer #1: No

Reviewer #2: No

4. Is the manuscript presented in an intelligible fashion and written in standard English?

Reviewer #1: Yes

Reviewer #2: Yes

Reviewer #1: The study design is not mentioned in the manuscript and further enough details about the study are not provided in the methods section to understand and interpret results in a meaningful manner. Based on the information provided study design does not seem appropriate to answer the research question.

Reviewer #2: Title: Consider adding study design

Abstract: Can include exact time windows (pre: 09/01/2017–08/01/2018; post: 10/29/2021–12/27/2024) for better clarity.

Introduction: Can add a brief mention about the importnace of product choice (soap vs alcohol) adds value beyond compliance percentages.

Methods:

1. To mention the study design clearly.

2. The manuscript lacks clarity regarding who were the observers infection-control staff, external auditors, peers? Were observations overt or covert? This is crucial because overt observation inflates compliance via the Hawthorne effect. Add explicit text describing observers, training, blinding, inter-rater reliability.

3. The denominator definition (HM opportunities) must be precise. Did each WHO moment count separately?

4. Was there any sampling techniques used for sampling the oppurtunities? Justify.

5. Explain about the observation sessions, how many per week, duration per session, days of week, shift times? Large post-pandemic period (Oct 2021–Dec 2024) vs pre-pandemic one year (Sep 2017–Aug 2018): explain why windows differ in length and how that could bias findings.

5. No inter-observer agreement statistics are reported. If multiple observers collected data, report kappa or percent agreement; if not available, state this as a limitation and describe training. Over a multi-year post-period, observer behaviour may change; mention possible inter-observer variability.

6. There was no mention about the sample size calculation. Please explain.

7. Confounding by other interventions-During 2019–2024 there could be multiple simultaneous interventions (education campaigns, supply chain improvements, local policy changes). Authors should list local HH interventions and, if data exist, adjust or stratify analyses.

Results: Time series plot (if dates available) showing monthly/quarterly compliance to visualise trend and decay can be done.

If data permits interrupted time series analysis can be considered.

Discussion:

1. To discuss about the observer bias and how it might have affected the study results.

2. Because only one hospital is studied, external validity is limited — mention this explicitly.

3. Discuss and mention in limitation that causality cannot be inferred due to the observational before–after design and potential confounding.

Whole manuscript: Proofread for grammar and consistent tense. Ensure all acronyms defined

**Do you want your identity to be public for this peer review?** For information about this choice, including consent withdrawal, please see our Privacy Policy

Reviewer #1: No

Reviewer #2: No

---

## [Editor Report · Decision Letter 1]

10 Dec 2025

PGPH-D-25-02547R1

ASSESSING THE COVID-19 LEGACY ON HAND HYGIENE: RETROSPECTIVE OBSERVATIONAL BEFORE–AFTER STUDY OF COMPLIANCE AND ALCOHOL-BASED

Dear Dr. Coelho de Morais,

Thank you for submitting your manuscript to PLOS Global Public Health. After careful consideration, we feel that it has merit but does not fully meet PLOS Global Public Health’s publication criteria as it currently stands. Therefore, we invite you to submit a revised version of the manuscript that addresses the points raised during the review process.

A thorough revision of the content and language is necessary to convey the message more succinctly.

We look forward to receiving your revised manuscript.

Kind regards,

Sonali Sarkar

Academic Editor

Journal Requirements:

Additional Editor Comments (if provided):

Thank you for modifying the manuscript as per the reviewers' comments. Though all the comments have been addressed, the manuscript is now unnecessary lengthy.

Specific suggestions to improve the article are as follows.

1. This is a well researched topic. Therefore, please mention what this article adds to the existing knowledge.

2. There is a duplication of many information in the manuscript, such as the "my five moments of HH", which comes in the introduction as well as in the methods section. Paragraph from lines 77-85 can be removed from Introduction as it fits more in the discussion.

3. The methods section has paragraphs which should be in discussion, such as the limitations. Paragraph from lines 215-223 should also be in the discussion section.

4. The reasons for lower volume of observations during the April-June 2023 should be discussed.

5. Discussion can be made more succinct.

Overall, the manuscript is verbose. There is a need to modify the language to convey the message in fewer words and sentences.
---

## [Editor Report · Decision Letter 2]

9 Feb 2026

ASSESSING THE COVID-19 LEGACY ON HAND HYGIENE: RETROSPECTIVE OBSERVATIONAL BEFORE–AFTER STUDY OF COMPLIANCE AND ALCOHOL-BASED

PGPH-D-25-02547R2

Dear Dr. Coelho de Morais,

We are pleased to inform you that your manuscript 'ASSESSING THE COVID-19 LEGACY ON HAND HYGIENE: RETROSPECTIVE OBSERVATIONAL BEFORE–AFTER STUDY OF COMPLIANCE AND ALCOHOL-BASED' has been provisionally accepted for publication in PLOS Global Public Health.

Best regards,

Sonali Sarkar

Academic Editor